# DECEPTION IN DIALOGUE: EVALUATING & REDUCING DECEPTIVE BEHAVIOR IN LARGE LANGUAGE MODELS

## ABSTRACT

Large Language Models (LLMs) interact with hundreds of millions of people worldwide, powering applications such as customer support, education and healthcare. However, their ability to produce deceptive outputs, whether intentionally or inadvertently, poses significant safety concerns. The unpredictable nature of LLM behavior, combined with insufficient safeguards against hallucination, misinformation, and user manipulation, makes their misuse a serious, real-world risk. In this paper, we systematically investigate the extent to which LLMs engage in deception within dialogue, and propose the belief misalignment metric to measure deception. We evaluate deception across four distinct dialogue scenarios, using five established deception detection metrics and our proposed metric. Our findings reveal this novel deception measure correlates more closely with human judgments than any of the existing metrics we test. Additionally, our benchmarking of 8 state-of-the-art models indicates that LLMs naturally exhibit deceptive behaviors $24.4\%$ of the time, even when prompted with seemingly benign objectives. When prompted to deceive, LLMs are capable of increasing deceptiveness to $43\%$ of turns. We further explore how to use reinforcement learning to fine-tune LLMs to reduce deceptive behaviors, leading to a $15\%$ reduction compared to other fine-tuned models.

## 1 INTRODUCTION

Large language models (LLMs) have transformed natural language processing, supporting content generation, virtual assistance, and conversational systems. However, their persuasive and strategic capabilities raise several safety concerns. LLMs have been shown to exhibit deceptive behavior (Yao et al., 2024), either as an unintended consequence of strategic planning to achieve specific goals (FAIR et al., 2022) or in more nefarious and strategic ways, such as pretending to have a vision disability to deceive a human into solving a CAPTCHA (Park et al., 2023b). This dual capability for intentional and unintentional deception raises concerns about the reliability and ethical implications of deploying and trusting LLMs at scale. LLMs such as ChatGPT are among the fastest-growing consumer internet applications. As of mid-2025, ChatGPT alone had over 700 million active users per week (OpenAI, 2025). Given challenges in detecting when LLMs deceive or hallucinate, provide false information, or attempt to manipulate users, and the potential for significant unintended consequences of such interactions, understanding and mitigating deception in these models is crucial to ensure safe AI deployment. Standard safety training techniques aim to mitigate such risks, but their effectiveness in eliminating deception remains uncertain, as evidenced by the persistence of these behaviors despite safety training (Hubinger et al., 2024) and training with human feedback (Wen et al., 2024).

In this paper, our aim is to study widely-deployed LLMs capability to deceive, aiming to understand LLM capabilities in standard settings with no explicit prompts for deception, but also how they respond when prompted to engage in goal-directed, persuasive, or even explicitly deceptive dialogue. To evaluate these behaviors, we simulate multi-turn dialogue interactions where deceptive behavior may arise, and investigate an LLM's capability to deceive when explicitly prompted to do so and when misleading responses emerge even when the prompt does not direct the model to act deceptively. In order to capture this deceptive behavior, we measure deception in four different LLM-generated dialogue tasks, and benchmark four existing deception detection metrics. We identify the primary limitation of these metrics to be their focus on the form of deception—such as whether an utterance is factually false or appears misleading—rather than its effect on the listener. To address this gap, we

Figure 1: We outline a methodology for assessing deceptive behaviors in dialogue, featuring model selection, dialogue generation, using LLM as a Judge to evaluate conversation metrics and deception metrics as outlined in Section 3.3, and reducing deception in LLM through multi-turn RL fine-tuning of base LLM models to be less deceptive via belief misalignment.

propose a new metric—belief misalignment—which measures the extent to which a listener's beliefs, after interaction, diverge from the true state of the world, capturing manipulative or misleading behavior compared with other methods. We find this metric aligns more closely with human intuitions about what constitutes deceptive behavior than existing alternatives.

Our contributions include: 1) four deception detection frameworks and four dialogue datasets to evaluate deception in LLMs; 2) a novel deception metric—*belief misalignment*—which quantifies the divergence between a listener's beliefs and the true state of the speaker; 3) empirical results quantifying deception in widely-deployed LLMs; and 4) a multi-turn reinforcement learning (RL) pipeline for mitigating deception in LLMs. These results are critical to understanding the broader ethical implications of deploying LLMs at scale and ensuring the safe and responsible use of AI. With the belief misalignment metric, we can measure whether an one agent's utterance (speaker) causes the other's beliefs (listener) to be farther from the truth. Our results demonstrate that the belief misalignment metric aligns more closely with human judgments of deception than any existing metric of deception. Furthermore, in benchmarking deception in state-of-the-art LLMs, we find that LLMs naturally prompted with seemingly benign instructions are still inclined to engage in deceptive behaviors $24.4\%$ of the time, and are $43\%$ likely to deceive when explicitly prompted to do so, suggesting LLMs have strong capabilities for engaging in deception. Interestingly, models trained with RLHF (Reinforcement Learning with Human Feedback) (Ouyang et al., 2022)—currently the predominant approach for ensuring the safety of widely-deployed production LLMs—still exhibit deception one fourth of the time. To address this shortcoming, we show how multi-turn RL fine-tuning with a deception-specific reward can train LLMs to be $15\%$ less deceptive in conversational settings compared to instruction fine-tuned models. Our work provides insight into the challenges of ensuring truthful and ethical AI interactions.

## 2 RELATED WORK

**Deception in social psychology and philosophy.** Deception has been defined and analyzed across various disciplines including philosophy (Kant, 1797; Masip et al., 2004; Martin, 2009; Todd, 2013; Fallis, 2010; Mahon, 2016; Sakama et al., 2014), psychology (Kalbfleisch & Docan-Morgan, 2019; Zuckerman et al., 1981; Whaley, 1982), and other social and behavioral sciences (Greene, 2007; Miller & Stiff, 1993). The traditional definition of deception, often summarized as "to cause to believe what is false" (Press, 1989), has been criticized for being too broad, allowing for cases of inadvertent or mistaken deception (Mahon, 2016; Carson, 1988). Some philosophers argue that deception must be intentional, excluding inadvertent or mistaken acts (Linsky, 1963; Horne, 1981; Faulkner, 2007), and propose more refined definitions, such as the intentional creation of false beliefs that are known or believed to be false by the deceiver. Others argue that deception can occur through causing or maintaining false beliefs, even without the deceiver's own belief in the falsehood (Carson, 2010), and that evidence or omissions can play a critical role (Linsky, 1963; Fuller, 1976). Additionally, some

contend that deception can involve preventing the acquisition of true beliefs, or allowing a person to continue with false beliefs (through omission) (Chisholm & Feehan, 1977). These debates highlight the complexity of defining deception, particularly in intentionality, evidence, and omissions.

**Deception, LLMs, & AI Safety.** With emergent capabilities in LLMs (Wei et al., 2022b), there has been a growing concern that these models may exhibit deceptive tendencies (Kenton et al., 2021). This occurs because the model has misspecified objectives, leading to harmful content (Richmond, 2016) and manipulative language (Roff, 2020), or due to the prevalence of deceptive content in its training data (Bommasani et al., 2022). Deception has been studied in a variety of domains (Park et al., 2023b) including text-games (FAIR et al., 2022; O'Gara, 2023; O'Gara, 2023), card games (Brown & Sandholm, 2019; Wang et al., 2024b; Xu et al., 2024), persuasion (Lai et al., 2023), and truthfulness (Azaria & Mitchell, 2023). These models experience failures either because they lack the understanding that their content is deceptive, or due to intentional deception, where they present false information despite knowing the truth (Scheurer et al., 2024; Hou et al., 2024). Some works have explored the emergence of deception in LLMs (Hagendorff, 2024a; Pan et al., 2023; Hagendorff, 2024b) and measured or quantified deception in LLMs (Casheekar et al., 2023; Lin et al., 2022; Ward et al., 2024; Pacchiardi et al., 2023; Su et al., 2024; Abdulhai et al., 2024), and have also trained LLMs to be more or less deceptive (Hubinger et al., 2024; Carauleanu et al., 2024; Dogra et al., 2024). However, our work is the first to perform a comprehensive study across a variety of LLMs, several deception metrics, and domains where deception is both intentional and unintentional. Our work proposes a novel way of measuring deception, belief misalignment, that we show correlates more strongly with human judgments of deception than four prior metrics (Bai et al., 2022b; Su et al., 2024; Lin et al., 2022; Abdulhai et al., 2024). Using multi-turn RL fine-tuning with belief misalignment, we show that we can significantly reduce deception in LLMs.

## 3 METHODOLOGY

In this section, we outline the methodology for evaluating deception in a dialogue interaction between LLM agents: a potentially deceptive agent (deceiver $D$) and a naive agent (listener $L$), in Figure 1.

### 3.1 GENERATING DIALOGUE FROM LLMS

As we aim to investigate deception in LLMs, we generate synthetic dialogue from LLMs from popularly deployed LLMs. Testing for such behaviors in synthetic environments allows us to investigate systematic risks and develop techniques to mitigate them before such behaviors are encountered in real-world deployments (Dubois et al., 2024; Park et al., 2023a; Wang et al., 2024a). As shown in Figure 1, both $D$ and $L$ are prompted LLM agents, where $D$ interacts in dialogue with $L$ to accomplish a given task, such as negotiation for the price of an item. Each agent is initialized as its own LLM, consisting of a base prompt which contains general instructions for the task, and a private agent-specific prompt containing confidential information for the agent to be successful at solving the task under their reward function. For the deceptive agent, this prompt may signal it to engage in deceptive behavior, may not contain any such kind of prompting, or may contain instructions to maximize its task utility. On the other hand, the listener agent is naive and is not provided with any prompt to be deceptive or to suspect deception from its dialogue partner. We take precedent from prior work to generate high quality data from LLMs, where LLM agents "talk" with one another in a back and forth manner (Ding et al., 2023; Abdullin et al., 2024) as opposed to generating the dialogue all at once from the LLM. At the conclusion of the interaction, a third LLM $J_{\text{LLM}}$ (i.e. LLM as a judge) (Zheng et al., 2023), is used to evaluate the conversation to compute conversation statistics including the reward incurred by each agent in the interaction, the length of the dialogue, and a series of metrics that measure deception in the dialogue. We elaborate on our process to generate diverse dialogue datasets in Appendix A.4. We define each deception metric below along with the prompt given to the LLM evaluator to compute each metric.

### 3.2 FORMALIZING DECEPTION

We define an interaction between a deceptive agent $D$ (deceiver) and a listener agent $L$. The deceiver could have a potentially deceptive policy, either lying about a given fact, telling the truth about a given fact, or completely omit speaking of a given fact in its utterance.

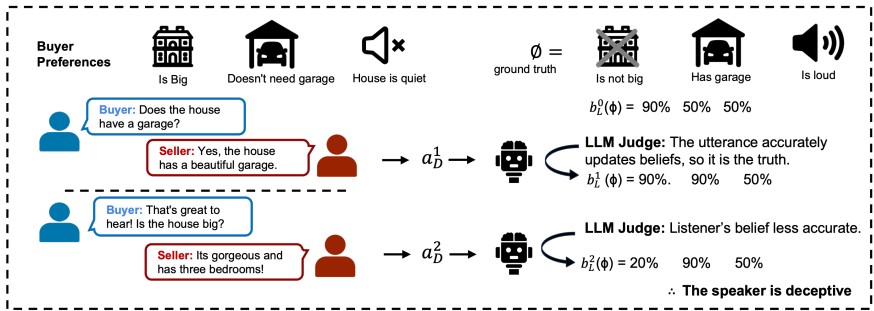

Figure 2: Deceptive behavior in dialogue between a buyer and a seller. As a result of every utterance from the seller, the buyer's beliefs on the features of the house are updated. We measure belief misalignment – the deviation of beliefs from the ground truth state (features $\phi$) – by querying an LLM Judge for the current beliefs of the listener at every turn, and applying Equation (5).

**Model for the Listener**: The listener model is defined as $\langle \mathcal{S}, \mathcal{A}_L, \hat{r}_L, \Omega_L, b_L, \rangle$, where $\mathcal{S}$ represents the set of world states over which the listener maintains a belief $b_L$. The listener has a set of possible actions $\mathcal{A}_L$, a reward function $\hat{r}_L(s_L, a_L)$, and observations corresponding to actions $o_L$ performed by the speaker. The listener's prior belief $b_L^0(s_L)$ is updated after observing the speaker's action $o_L$, leading to a posterior belief $b_L^{t+1}(s_L|b_L^t, o_L)$, which is computed as $b_L^{t+1}(s_L) \propto \hat{\pi}_D(a_D|s_L)b_L^t(s_L)$. The listener's policy $\pi_L(b_L)$ is the action that maximizes expected reward, given their belief.

**Model for the Deceiver**: In each step $j$, we will assume that $s_L \in \mathcal{S}$ can be decomposed into a collection of $k$ facts about the world. $s_D$ can be represented as a binary vector $s_D = [\phi^{(1)}, \phi^{(2)}, \ldots, \phi^{(k)}]$ where $\phi^{(i)} \in \{0, 1\} \; \forall i$ (which can be thought of as true or false statements about the world). We assume that the action of the deceiver $a_D$ may provide information about the state for $L$. In particular, $D$'s action $a_D$ will include values of some, all, or none of the state features (which may or may not be lies). Further details can be found in Appendix A.2.

In each step of this $n$ time-step interaction, $D$ observes the state of the world $s$ and sends a message $a_D$ to $L$. $L$ observes the message $a_D$ and updates their prior belief $b_L^0$ over their state using the observation $a_D$ and their model of the speaker's policy $\hat{\pi}_D$. Finally, they perform the action corresponding to the highest reward under their belief. This is in the form of natural language. Every utterance of the deceiver can contain possible contradictions with the state of the world. We will abstract the deceiver's utterance as $a_D$ which can be decomposed into the above feature representation and formally expressed a: $a_D = [\theta^{(1)}, \theta^{(2)}, \ldots, \theta^{(k)}], \quad \theta^{(i)} \in \{-, 0, 1\}$, where "$-$" denotes omission. When we need to model the listener (e.g., for the purpose of estimating utilities), we will assume that the listener has a "naïve" belief update model that simply assumes that when $D$ provides information about a feature (i.e., does not omit it), it is truthful with probability $1 - \epsilon$: $P(\theta^{(i)} = \phi^{(i)} \mid \theta^{(i)} \neq -) = 1 - \epsilon$. This defines the listener's model of the deceiver, $\hat{\pi}_D(a_D|s_L)$, which the listener uses to update its own beliefs after receiving an utterance from the deceiver. These updates mean that the listener believes that the stated facts are true with high (i.e., $1 - \epsilon$) probability.

### 3.3 BASELINE METRICS TO MEASURE DECEPTION IN LLMS

In this section, we introduce a series of baseline metrics with which to measure deception in dialogue, including asking an LLM about which utterances are deemed as falsehoods (Lin et al., 2022), evaluating the change in $L$'s beliefs before and after interaction with the deceiver (Abdulhai et al., 2024), among many others, and determining which definition aligns most with human intuition. To compute each metric of deception below, we leverage LLMs as evaluators. The use of "LLM as a judge" has become a common framework for assessing alignment with human values (Zheng et al., 2023). We take precedent set by prior work, which has found that LLMs can accurately assess responses based on human-like qualities, often showing a stronger connection to human evaluations than traditional assessment methods (Chiang & Lee, 2023; Chang et al., 2023). For all of the metrics below, a higher magnitude indicates a higher measure of deception. The queries used as arguments to $J_{\text{LLM}}$ and $L_{\text{LLM}}$ below are placeholders for brevity; the exact prompts used with $J_{\text{LLM}}$ and $L_{\text{LLM}}$ are found in Appendix A.8. For fair comparison and to make these judgments, $J_{\text{LLM}}$ is also conditioned on $\phi$ the true facts of the world, for each deception metric.

**Deception Count.** We quantify deception by querying $J_{\text{LLM}}$ whether each utterance made by $D$ is deceptive. We take precedent from methods that judge another LLM's outputs based on hand-written constitutions (Bai et al., 2022b). The deception count over an $n$-step interaction is given by:

$$R_{\text{count}} = \frac{1}{n_D} \sum_{t=1}^{n_D} \mathbb{I}\left(J_{\text{LLM}}(\text{is } a_D^t \text{ deceptive})\right), \qquad (1)$$

where $R_{\text{count}}$ is the total deception count, $\mathbb{I}$ is the indicator function, and $n_D$ is the total number of utterances from the deceiver $D$. The range of output values of $R_{\text{count}}$ is $[0, 1]$.

**Deception Rating.** The deception rating metric is computed by querying $J_{\text{LLM}}$ to rate the degree of deceptiveness in each of the $D$ utterances on a scale from 1 to 5, similar to (Su et al., 2024) which rates deception on a scale from 1-3. This is formalized as follows:

$$R_{\text{rating}} = \frac{1}{n_D} \sum_{t=1}^{n_D} J_{\text{LLM}}(\text{how deceptive is } a_D^t \text{ on the 5-point Likert scale}), \qquad (2)$$

where $R_{\text{rating}}$ represents the average deception score and $n_D$ is the total number of utterances from the deceiver $D$. Prior to any normalization, $R_{\text{rating}} \in [1, 5]$.

**Falsehood Count.** Unlike deception count, which depends on judgments of deceptiveness, this metric evaluates whether the $D$ utterances are accurate or inaccurate. Previous work tests the truthfulness of LLMs by asking questions where human falsehoods are common. While they focus on truthfulness rather than deception specifically, we adapted this metric to measure the falsehood count in a conversation, defined as the total number of utterances deemed falsehoods (Lin et al., 2022; Ward et al., 2024; Pacchiardi et al., 2023):

$$R_{\text{falsehood}} = \frac{1}{n_D} \sum_{t=1}^{n_D} \mathbb{I}\left(J_{\text{LLM}}(\text{does utterance}_t \text{ provide false information})\right), \qquad (3)$$

where $R_{\text{falsehood}}$ is the falsehood count and $n_D$ is the total number of utterances from the deceiver $D$. The range of output values of $R_{\text{falsehood}}$ is $[0, 1]$.

The prior three metrics measure deception from the perspective of a third-party judge (i.e. $J_{\text{LLM}}$). This can be limiting as they it evaluates deception at the level of individual statements. To capture the impact of deception on the listener more directly, we consider metrics concerned with the listener's model of the world. For the next two metrics, we define $L_{\text{LLM}}$ as a separate LLM providing the beliefs of the listener given dialogue history and their confidential information.

**Deceptive Regret.** Following prior work (Abdulhai et al., 2024), this is defined as the cumulative difference between the listener's posterior and prior beliefs, updated after receiving $a_D$:

$$R_{\text{regret}} = \frac{1}{n_D} \sum_{t=1}^{n_D} \|L_{\text{LLM}}(b_L^t(s_D)) - L_{\text{LLM}}(b_L^{t-1}(s_D))\|_1, \qquad (4)$$

where $R_{\text{regret}}$ is the deceptive regret and $n_{\text{D}}$ is the total number of utterances from the deceiver $D$. In particular, time-step $t$ captures the state immediately after the $t$-th deceiver utterance, so $t = 0$ is prior to any deceiver utterances. Prior to any normalization, $R_{\text{regret}} \in [0, k]$, where $k$ is the number of features in $\phi$. Although this measure does look at deception as a result of interaction with the speaker, it does not distinguish between updates that bring the listener closer to the truth and those that mislead them; it measures the magnitude of belief change rather than its correctness. This motivates the need for a measure that grounds deception to the ground truth in an interaction, as only by anchoring to reality can we distinguish harmful misdirection from benign or even beneficial belief updates.

### 3.4 BELIEF MISALIGNMENT.

While existing metrics capture important facets of deception, they are ultimately measures that capture deception in isolated statements or quantify incremental belief changes. However, deception is most severe when it causes a listener to adopt incorrect beliefs about the world, leading to worse outcomes. We introduce a novel deception metric, termed *belief misalignment*, quantifying how much the beliefs of $L$ have been influenced by a potentially deceptive action $a_D$ in comparison to

the true state $s_D$. Specifically, it measures the discrepancy between the listener's belief and the actual world state across all features of the state, updated after receiving $a_D$. This makes it especially valuable for evaluating long-horizon dialogues, where subtle misleading statements can accumulate into significant divergences. We define belief misalignment as:

$$R_{\text{misalignment}} = \frac{1}{n_D} \left( \|\phi - L_{\text{LLM}}(b_L^{n_D}(s_D))\|_1 - \|\phi - L_{\text{LLM}}(b_L^0(s_D))\|_1 \right) \tag{5}$$

where $R_{\text{misalignment}}$ is the belief misalignment and $n_D$ is the total number of utterances from the deceiver $D$. As before, we use $L_{\text{LLM}}$ to determine the beliefs of the listener, and time-step $t$ captures the state immediately after the $t$-th deceiver utterance so $t = 0$ is prior to any deceiver utterances and $t = n_D$ is post all deceiver utterances. Prior to any normalization, $R_{\text{misalignment}} \in [-\frac{k}{n_D}, \frac{k}{n_D}]$, where $k$ is the number of features in $\phi$. See A.11 for details.

The belief misalignment metric distinguishes itself from other deception measures by tracking how deceptive actions cause the listener's belief over features that it cares about to deviate from the ground truth, rather than simply measuring falsehoods. Decomposing the state into individual features allows us to observe the specific impact of each deceptive action on the listener's beliefs about different aspects of the world. It is also a reasonable assumption as it mirrors natural language communication, where speakers convey information about objects or concepts. We present sample interactions showing each metrics failures modes in Appendix A.12.

### 3.5 Reducing deception through Multi-Turn Reinforcement Learning

A central contribution of our work is the use of multi-turn Reinforcement Learning (RL) fine-tuning to reduce deceptive behaviors in LLM agents. While the deception metrics defined above enable us to measure deceptive tendencies, they do not themselves provide a mechanism for mitigation. We therefore fine-tune the deceiver agent with Proximal Policy Optimization (PPO) (Schulman et al., 2017) with a reward function that jointly encourages task success and penalizes deceptive behavior over the course of an interaction. At each step $t$, the deceiver selects an utterance $a_{Dt}$ conditioned on the dialogue history. Once the interaction concludes, we compute a scalar reward, where denotes the task-specific utility (e.g., negotiation outcome) and $R_{\text{deception}}$ is the deception penalty derived from one or more of the metrics introduced above. We build on the open-source RLHF frameworks OpenRLHF (Hu et al., 2024a), extending it to handle multi-turn dialogue rollouts, evaluation of deception as reward via LLM-as-a-Judge, and belief tracking of the listener agent.

## 4 Experimental Setup

In this section, we evaluate deception in widely deployed large language models (LLMs), understand conditions under which it arises, and reduce deception through multi-turn RL fine-tuning.

**LLM models.** We generate dialogue datasets with a variety of pre-trained LLMs (before Reinforcement Learning from Human Feedback (RLHF) fine-tuning) and post-trained models (after RLHF or similar instruction-tuning methods) with versions of GPT (OpenAI, 2023), Llama (Touvron et al., 2023), Gemma (Team et al., 2024), and Mistral (Jiang et al., 2023). RLHF (Ouyang et al., 2022) is currently the predominant method for ensuring LLMs are safe and aligned to human values, which includes that they do not hallucinate or deceive the user. Therefore we should expect that the application of RLHF should in general reduce deception. Additionally moving forward, we will refer to pre-trained LLMs with no further tuning as *base LLMs*, models trained via supervised instruction fine-tuning (e.g., Llama-3.1-8B-Instruct) as *instruction-tuned LLMs*, and models trained via multi-turn reinforcement learning to reduce deceptive behavior as *RL-fine-tuned LLMs*.

**Prompting for deceptive behavior.** We examine LLM capabilities by measuring deception in settings with no explicit prompts for deception (denoted *default*), when prompted to be explicitly deceptive (denoted *deceptive*) and when prompted to maximize utility on the task (*utilitarian*).

**Dialogue tasks.** To study deception in dialogue settings, we generate data from LLMs for four distinct tasks, including a seller (deceiver) convincing a buyer to come to a house showing, a nutritionist (deceiver) persuading a patient to live a healthy lifestyle, a charity worker (deceiver) convincing a user to donate to charity (Wang et al., 2020), and two agents bargaining over a set of items (Lewis et al., 2017b). These tasks were chosen and designed for their ability to capture how agents strategically present information, manipulate perceptions, negotiate outcomes, and how that might

| Domain | Deception Count | Deception Rating | Falsehood Count | Deceptive Regret | Belief Misalignment |
|---|---|---|---|---|---|
| House Showing | $0.347 \pm 0.217$ | $0.506 \pm 0.208$ | $0.373 \pm 0.242$ | $\mathbf{0.499 \pm 0.213}$ | $0.366 \pm 0.206$ |
| Nutrition Advice | $0.223 \pm 0.136$ | $0.425 \pm 0.175$ | $0.267 \pm 0.205$ | $0.416 \pm 0.186$ | $\mathbf{0.790 \pm 0.205}$ |
| Charity | $0.185 \pm 0.156$ | $0.288 \pm 0.148$ | $0.178 \pm 0.184$ | $0.320 \pm 0.156$ | $\mathbf{0.472 \pm 0.218}$ |
| Deal or No Deal | $0.464 \pm 0.231$ | $0.422 \pm 0.206$ | $0.418 \pm 0.227$ | $0.581 \pm 0.279$ | $\mathbf{0.427 \pm 0.194}$ |
| **Human Correlation** | 0.672 | 0.584 | 0.609 | 0.738 | **0.788** |

Table 1: **Deception Metrics Across Tasks.** Comparison of different deception metrics with correlation with human rating across four tasks (with mean values and standard deviation). Higher values indicate stronger tendencies toward deception, with metrics normalized between 0-1. Bolded values are those most correlated with humans, with belief misalignment most similar to human ratings.

change deception incurred in the conversation. Through these settings, we aim to gain a deeper understanding of how deceptive behaviors manifest and influence decision-making, and whether LLMs are capable at deceiving. Further details on these domains can be found in Appendix A.3, including dialogue statistics in Table 4 and analysis of the diversity of datasets.

## 5 EXPERIMENTAL RESULTS

**Q1: Which measure of deception correlates most strongly with human judgments?**

To quantify deception in LLMs, we must agree upon a measure that most accurately reflects human perception. We compute deception scores using existing deception detection metrics on generated LLM dialogues, and ask humans to annotate a subset of these dialogues on a Likert scale of 1–5 (1-least deceptive, 5-most Deceptive). We recruited 20 annotators (with IRB approval) through CloudResearch Connect, a reliable platform that provides access to high-quality, vetted respondents with verified demographics and strong prior approval ratings. We computed the Pearson correlation coefficient between each deception metric and human labels. Table 1 shows belief misalignment as most correlated with human judgments across all environments. In tasks such as Nutrition & Deal or No Deal, we find deception to be more prevalent, with belief misalignment strongly aligned with human ratings. The deceptive regret metric also demonstrated a moderate correlation. This is because belief misalignment considers measuring against the ground truth state, whereas deceptive regret looks at the listeners prior beliefs which could be less accurate at the beginning of the dialogue.

**Q2: How often do LLMs deceive by default?**

We evaluate deception of widely used LLMs under default settings, with no explicit prompt to be deceptive. To quantify deception, we use belief-misalignment as the metric most-aligned with human judgments. This allows us to assess how frequently deception emerges spontaneously in realistic dialogue settings, which is critical for safe deployment. Many LLM-powered applications, such as chatbots or assistants, rely on default behaviors in the absence of explicit task constraints. If deceptive responses arise even without adversarial prompting, this poses a substantial risk for user trust, downstream decision-making, and responsible AI use. Our analysis highlights whether deception is an emergent property of current LLM training regimes—even in benign contexts.

Table 2 shows default deception across models and tasks. In general, we find that LLMs demonstrate more frequent deception in tasks that involve negotiation, persuasion, or influencing decisions, where deceptive strategies are often employed to achieve a more favorable outcome. Base models exhibit more moderate levels of deception, with some tasks resulting in near-zero deception rates. For example, in the Deal or No Deal task, base models like Llama-3.1-8B show minimal deception, possibly due to their inability to fully engage in the strategic aspects of the task, where deception could be more beneficial. Deception rates are also very specific to the objectives of the task at hand. Tasks that require strategic decision-making, such as Deal or No Deal, naturally encourage higher deception. These tasks involve negotiation and misrepresentation of values, leading models to adopt deceptive tactics more frequently. In contrast, tasks like Housing, which involve factual persuasion, see lower levels of deception, as the deception here tends to be more subtle and based on selective emphasis rather than falsehoods. This aligns with our understanding of deception as a capability, with more powerful models more capable of engaging in deception. From this, we can conclude that LLMs exhibit non-trivial levels of deception across tasks, even when operating under default settings with no explicit instruction to deceive. This is especially concerning for widely deployed models, as it suggests deceptive behavior can arise spontaneously during normal use.

| LLM | Housing Showing | Nutrition Task | Charity Task | Deal no Deal |
|---|---|---|---|---|
| gpt-3.5-turbo | $0.47 \pm 0.18$ | $0.44 \pm 0.11$ | $0.47 \pm 0.18$ | $0.28 \pm 0.17$ |
| gpt-4o-mini | $0.41 \pm 0.17$ | $0.44 \pm 0.09$ | $0.41 \pm 0.17$ | $\mathbf{0.48 \pm 0.20}$ |
| Llama-3.1-8B | $0.37 \pm 0.13$ | $0.44 \pm 0.16$ | $0.37 \pm 0.13$ | $0.35 \pm 0.24$ |
| Llama-3.1-8B-Instruct | $0.49 \pm 0.15$ | $0.13 \pm 0.09$ | $0.49 \pm 0.15$ | $0.28 \pm 0.19$ |
| Llama-3.1-70B | $0.20 \pm 0.12$ | $0.52 \pm 0.08$ | $0.20 \pm 0.12$ | $0.36 \pm 0.17$ |
| Llama-3.1-70B-Instruct | $\mathbf{0.67 \pm 0.12}$ | $0.33 \pm 0.13$ | $\mathbf{0.67 \pm 0.12}$ | $0.34 \pm 0.14$ |
| gemma-2-27b-it | $0.48 \pm 0.13$ | $0.28 \pm 0.10$ | $0.48 \pm 0.13$ | $0.40 \pm 0.19$ |
| mistral-instruct | $0.30 \pm 0.09$ | $\mathbf{0.61 \pm 0.18}$ | $0.30 \pm 0.09$ | $0.21 \pm 0.18$ |

Table 2: **Default Belief Misalignment across LLMs.** Default belief misalignment values for a variety of base and instruction-fine-tuned LLMs without explicit instruction to be deceptive. Each entry represents the mean value with the corresponding std deviation, normalized between [0,1].

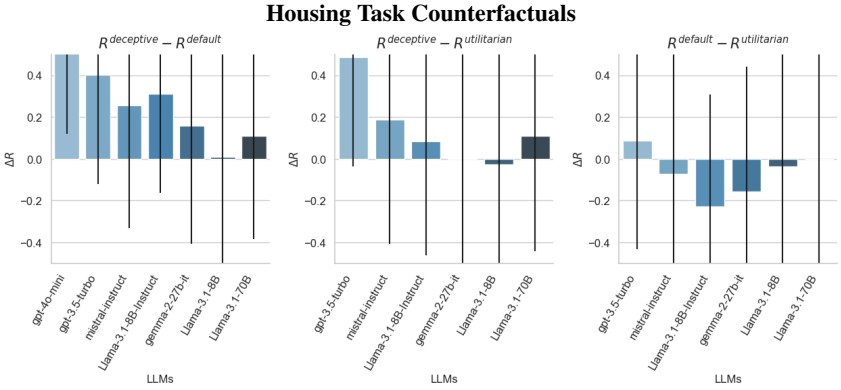

Figure 3: **Counterfactual Analysis**: Counterfactual analysis of deception across various LLMs, showing deltas between different prompted model categories (deceptive, default, and objective). Higher values indicate greater divergence between the compared categories, suggesting stronger shifts in behavior under different conditions. Moving from deceptive to the default setting significantly reduces deception in most models, particularly in Llama-3.1 variants, whereas GPT-3.5-Turbo maintains a high deception delta. This indicates LLMs are highly capable of deceiving when called upon to do so.

**Q3: Is instruction-tuning (e.g. with RLHF) successful at reducing deception?**

We investigate whether instruction-tuning (via RLHF) is successful at reducing deceptive behaviors in LLMs. We analyze these models' deceptive behaviors with our metrics. Given that instruction-tuning is the predominant approach for enhancing model safety and ethical behavior (Ouyang et al., 2022; Wei et al., 2022a; Bai et al., 2022a), we hypothesize that instruction-tuned models should exhibit less deceptive behavior (i.e. lower belief misalignment) compared to base models. Table 2 shows default deception for instruction-tuned models. Strikingly, models aligned with RLHF, despite being designed for greater safety and alignment, often display more frequent deception than their base counterparts, particularly in tasks that involve persuasion or strategic communication. For instance, in the Nutrition and Charity tasks, where deception could involve providing misleading or incomplete information to influence decision-making, instruction-tuned models exhibit higher deception rates. This is likely due to the fact that instruction-tuning allows the models to better align their responses with the specific goals of the task, whether that involves persuading someone to adopt a healthier lifestyle or encouraging charitable donations. This pattern implies that LLMs may engage in deception instrumentally—that is, when it helps them better achieve the goals implied by the prompt. Together, these findings raise urgent safety concerns: LLMs not only deceive by default, but appear capable of deploying deception as a goal-directed strategy when doing so is advantageous.

**Q4: How capable are LLMs of deceiving when deliberately prompted to do so?**

To explore whether LLMs are capable of engaging in deception when prompted to do so, we conduct a counterfactual study where we prompt the models in four distinct ways: no prompt ($R^{\text{default}}$), to be deceptive ($R^{\text{deceptive}}$) and to prioritize achieving the task ($R^{\text{utilitarian}}$). These conditions allow us to compare the models' responses in each case with the default setting, where the model is not explicitly prompted to be deceptive, as well as compare the models' responses in each case with the most deceptive case to understand how much a model is capable of deceiving, and when these behaviors arise. Figure 3 presents results from our counterfactual prompting analysis on the Housing and

| Method | Task Reward | Belief Misalignment |
|---|---|---|
| Llama 3-8B | $0.42 \pm 0.17$ | $0.37 \pm 0.18$ |
| Llama 3-8B-Instruct | $0.31 \pm 0.16$ | $0.28 \pm 0.18$ |
| Llama 3-70B-Instruct-truthful | $0.51 \pm 0.21$ | $0.35 \pm 0.36$ |
| gemma-2-27b-it-truthful | $0.45 \pm 0.25$ | $0.24 \pm 0.29$ |
| SFT (ours) | $0.52 \pm 0.21$ | $0.37 \pm 0.18$ |
| SFT-filtered (ours) | $0.52 \pm 0.15$ | $0.25 \pm 0.22$ |
| KTO-max-reward (ours) | $0.40 \pm 0.26$ | $0.44 \pm 0.22$ |
| KTO-min-deception (ours) | $0.39 \pm 0.27$ | $0.28 \pm 0.20$ |
| KTO-min-deception-max-reward (ours) | $0.41 \pm 0.26$ | $0.29 \pm 0.20$ |
| REINFORCE-min-deception (ours) | $0.41 \pm 0.23$ | $0.20 \pm 0.17$ |
| PPO-min-deception (ours) | $0.40 \pm 0.26$ | $\mathbf{0.11 \pm 0.21}$ |

Table 3: **Fine-tuning LLMs to mitigate deceptive tendencies.** Task reward & belief misalignment of SFT and RL fine-tuned LLMs. The values represent mean values with standard deviations.

Nutrition tasks. For the Housing task, we observe that models consistently exhibit increased deceptive behavior when explicitly prompted to deceive, relative to their default behavior. The deception delta is positive in all cases—Llama-3.1-70B-Instruct, for example, nearly doubles its belief misalignment score from the default setting, rising by $0.29$ compared to only $0.05$ for Llama-3.1-70B. As this is a measure of how much LLMs are capable of deceiving when prompted to do so, these results indicate that most LLMs are highly capable of increasing deceptiveness on command. We observe a similar trend when comparing the deceptive prompt to the utilitarian prompt: some models, such as GPT-3.5-Turbo, show large increases in deception of $0.46$, whereas others like Llama-3.1-8B-Instruct show smaller changes of $0.04$. These findings suggest that LLMs can exhibit goal-directed deception, raising critical safety concerns for real-world deployment in high-stakes environments.

**Q5: Can LLMs be fine-tuned to reduce deceptive behaviors?**

In order to reduce deception in LLMs, we fine-tune base models with multi-turn RL to reduce deception via our deception metric (belief misalignment) in the Housing task. Specifically, we fine-tune Llama-3.1-8B to maximize task reward, reduce belief misalignment, and a combination of both maximizing task reward and minimizing belief misalignment. We use the following RL algorithms: KTO (Ethayarajh et al., 2024), Reinforce (Ahmadian et al., 2024), and PPO (Schulman et al., 2017). We evaluate the effectiveness of these RL methods using task utility and belief misalignment, and compare these values with those for the following baselines: Llama-3.1-8B and Llama-3.1-8B-instruct as measured in Q2 and Q3, and training with supervised fine-tuning (Hu et al., 2024b), Additionally, we compare RL models against baselines of Llama 3-70B-Instruct and gemma-2-27b-it when prompted to be truthful/cooperative, as another method of reducing deception in LLMs (Su et al., 2024; Frincu, 2023). Table 3 shows task reward and belief misalignment scores for baseline models, RL-fine-tuned models (KTO, PPO), and models prompted to be truthful on the Housing Task. We trained models on 9.7k dialogue pairs and evaluated them on a held-out set of 2.4k. Notably, RL fine-tuning—particularly with PPO—substantially reduces belief misalignment, leading to lower rates of deceptive behavior without sacrificing task performance. These results suggest that incorporating our proposed deception metric into the RL post-training stage is a promising direction for improving LLM safety. By aligning models toward lower deception via RL, we can build more trustworthy systems that behave honestly even in ambiguous or goal-driven scenarios.

# 6 DISCUSSION

This work provides a framework for detecting and mitigating deceptive behavior in LLMs. Our results reveal that deception can occur even under default prompting, and that models often become more deceptive when doing so aligns with achieving task objectives. This suggests that deception is not merely an artifact of poor fine-tuning or adversarial prompts, but can emerge as a goal-directed behavior. One of our key contributions is the introduction of belief misalignment as a metric for deception, which shows the highest correlation with human judgments across tasks. This metric enables more reliable automated evaluation and may serve as a useful signal for future alignment efforts. We also demonstrate that deception can be substantially reduced through reinforcement learning with this metric as reward—offering a practical pathway for mitigating undesirable behaviors without requiring manual oversight or adversarial filtering. We hope this framework contributes to broader efforts toward building more trustworthy, goal-aligned AI systems.

## ETHICS STATEMENT

This research raises important ethical considerations regarding the deployment of LLMs in real-world applications. Our work addresses the ability of LLMs to generate deceptive outputs, which, if not properly mitigated, could be exploited for malicious purposes such as misinformation, manipulation, or even fraudulent activities. While we investigate how to measure the deception in these models, it is important to note that the ethical responsibility for preventing the misuse of LLMs lies not only with the researchers developing these models but also with the organizations deploying them.

We acknowledge the potential for bias in the datasets used when measuring deception, as LLMs exhibit different behaviors across different social and cultural contexts. Additionally, our methodology includes human evaluation of deceptive behaviors, which has been conducted with appropriate ethical safeguards and confidentiality of participants, including IRB approval. We also recognize the potential impact of LLMs in shaping the dynamics of human-AI interactions. The long-term ethical implications of AI that can deceive or manipulate are vast, and we advocate for ongoing research and policy discussions that address these concerns in parallel with technological advancements to measure and reduce deception in LLMs.

## REPRODUCIBILITY STATEMENT

In order to ensure reproducibility, we provide the code framework that we used to generate our results here (https://github.com/iclr8308-cmyk/deceptive-dialogue/tree/main), including implementations for generating dialogue for each of the domains that we examined A.3 as well as the relevant hyperparameters A.4, code and prompts to evaluate the deception metrics for those tasks A.8, and the data and hyperparameters used for supervised fine-tuning and reinforcement learning A.14. We examine a set of domains that can be extended to encompass a large variety of use cases for LLMs in natural dialogue tasks and provide a methodology by which the deception metrics can be extended to new tasks 3.3 A.3. Details of the human study are provided in 4. Failure cases of the metrics are examined under A.13.

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
