# OpenReview forum: "Deception in Dialogue: Evaluating and Mitigating Deceptive Behavior in Large Language Models"
_ICLR.cc/2026/Conference — Submitted to ICLR 2026_

### Official Review · Reviewer_GGEr · 2025-10-26

**Soundness:** 3
**Presentation:** 3
**Contribution:** 3
**Rating:** 6
**Confidence:** 2

**Summary:**

The paper systematically studies deceptive behavior in dialogue settings for LLMs and proposes a new metric, belief misalignment, which measures how far a listener’s beliefs deviate from the true state of the world after interaction. Across four dialogue tasks, the authors compare existing deception metrics against belief misalignment and find the new metric aligns best with human annotations. Benchmarking shows that with seemingly benign prompts, LLMs still exhibit deception at a nontrivial rate (24.4%), rising to 43% when explicitly prompted to deceive. The authors then apply multi-turn reinforcement learning to jointly optimize task reward and an anti-deception objective, reducing deceptive behavior by about 15% at the dialogue level.

**Strengths:**

1.    Important, practically relevant problem. The paper targets real-world risks of deception/misdirection in LLMs, uses multi-task, multi-model evaluation, and offers contributions with clear safety and ethical significance.

2.    Systematic experiments and ablations. The study contrasts default, explicitly deceptive, and utility-maximizing prompting regimes; and compares base, instruction-tuned, and RL-tuned models, revealing how goal-directed prompting increases deception and how effects vary by task.

3.    Clear, reproducible mitigation path. The multi-turn RL fine-tuning with an anti-deception reward (PPO best) meaningfully lowers belief misalignment without materially harming task reward; the setup appears replicable and extensible.

**Weaknesses:**

1.    Dependence on LLM-as-Judge and potential circularity. Several metrics (including belief misalignment) rely on LLM evaluators to infer listener beliefs or label deception, importing evaluator bias/drift. Beyond human-correlation checks, the paper should add cross-evaluator robustness (different architectures/scales) and adversarial/attack tests to ensure the judge is not easily swayed.

2.    Idealized truth modeling and listener assumptions. Decomposing world state into k facts and assuming a “naïve update” listener aids formalization but abstracts away messy real-world issues (open-domain truth, uncertainty, multi-source evidence, value vs. fact). Testing on noisier, semi-structured factual tasks (e.g., medical/travel advice) and reporting failure modes would strengthen claims.

3.    Trade-offs with truthfulness/harmfulness not deeply analyzed. While RL reduces deception and preserves task reward, the paper provides limited analysis of how reducing deception interacts with other axes (truthfulness, politeness, persuasive efficacy). Instruction-tuned models’ higher deception on some tasks is mostly explained post hoc; targeted interventions (vary prompt strength/reward shaping) would be informative.

**Questions:**

see Weaknesses

---

> ### Author Response · Authors · 2025-11-21
>
> We appreciate your valuable feedback and your recognition of our work as addressing a “practically relevant problem.” In response to the main questions raised in your review, we have expanded the paper by: (1) adding an experiment that compares our LLM-judge metrics against multiple alternative LLM-judge models to assess cross-evaluator robustness, (2)  extending our experimental evaluation of deception to include an additional domain with hard to define ground truth (3) incorporating open-ended conversational quality assessments using AlpacaEval-2 along with a fine-grained dialogue evaluation inspired by the FED metric.
>
> 1. **“Dependence on LLM-as-Judge and potential circularity [...] the paper should add cross-evaluator robustness (different architectures/scales) and adversarial/attack tests to ensure the judge is not easily swayed”**
>
> We thank you for raising this important point. Following your suggestion, we have conducted an experiment to measure cross-evaluator robustness across two distinct judge models: gpt-5o-mini and Llama-3.1-70B-Instruct. We have recomputed our deception metrics on 80 conversations across two domains (Housing & Deal or No Deal Task), with an equal distribution of conversations where the agent is prompted with the default case (no explicit prompting to be deceptive) and deceptive prompting.  Our goal is to evaluate whether our deception metrics remain stable when the judge model changes, which would indicate that the metrics are not overly sensitive to any one model’s biases or priors. We report the Spearman correlations in Table 1 below for the Housing Task and Table 2 for the Deal or No Deal Task. As shown, we consistently observe strong positive correlations between judges. In the Housing task, we see belief misalignment shows perfect agreement across evaluators, indicating highly stable judgment of listener beliefs. The Deal or No Deal task similarly shows high cross-model correlation, with correlations typically between 0.70 and 1.00. We will include these additional results in our revised paper.
>
> ### Table 1: Housing Correlations between LLM Judges
> | Model Pair                         | deception_count| deception_score | falsehood_count | deceptive_regret | belief_misalignment |
> |-----------------------------------|----------------------|----------------------|----------------------|------------------|---------------------|
> | gpt-4o-mini – gpt-5o-mini         | 0.816                | 1.0                  | 0.681                | 1.0              | 1.0                 |
> | gpt-4o-mini – Llama-3.1-70B-Instruct | 0.820                | 1.0                  | 0.605                | 1.0              | 1.0                 |
> | gpt-5o-mini – Llama-3.1-70B-Instruct  | 0.570                | 1.0                  | 0.548                | 1.0              | 1.0                 |
> ### Table 2: Deal or No Deal Correlations between LLM Judges
>
> | Model Pair                         | deception_count | deception_score | falsehood_count | deceptive_regret | belief_misalignment |
> |-----------------------------------|-------------------------|-------------------------|-------------------------|----------------------|-------------------------|
> | gpt-4o-mini – gpt-5o-mini         | 1.000                   | 1.000                   | 1.000                   | 1.000                | 0.985                   |
> | gpt-4o-mini – Llama-3.1-70B-Instruct  | 0.892                   | 0.710                   | 0.841                   | 1.000                | 0.802                   |
> | gpt-5o-mini – Llama-3.1-70B-Instruct | 0.892                   | 0.699                   | 0.841                   | 1.000                | 0.779                   |
>
> 2. **Idealized truth modeling and listener assumptions [...]**
>
> To address your question, we have extended our experimental evaluation of deception to include an additional domain with semi-structured data, details of which are found in response #1 for Reviewer NC2x. In more open-ended or subjective settings where ground truth facts are not explicitly defined as shown in our new experiment, belief misalignment can instead be anchored to the user’s own stated or inferred goals, and becomes a measure of how far the Assistant steers the user away from their expressed intentions, rather than from an objective world state. More generally, without a ground truth, belief misalignment can be defined by whether the deceptive speaker agent believes one thing yet steers the listener toward a different belief, even if the deceptive speaker agent's underlying belief is imperfect or uncertain.

---

> ### Author Response · Authors · 2025-11-25
>
> 3. **“Trade-offs with truthfulness/harmfulness not deeply analyzed. While RL reduces deception and preserves task reward, the paper provides limited analysis of how reducing deception interacts with other axes (truthfulness, politeness, persuasive efficacy).”**
>
> We thank the reviewer for their thoughtful question. To address your comment, we have provided 2 additional evaluations with open-ended conversational quality assessment through AlpacaEval-2 [1] (an LLM-based automatic evaluator for instruction-following models) and fine-grained evaluation of dialog inspired by the FED metric [2]. On AlpacaEval-2, our model achieves a win rate of 17.06%, roughly matching similarly sized models (range of 10.59%-20.18%) as shown in Table 3 below. This suggests that our method does not harm general instruction-following.
>
> Additionally, we perform evaluation via LLM-as-a-judge (with gpt-4o-mini) on the same set of eighteen fine-grained dialog qualities as FED [2], which contains the other axes noted by the reviewer. Given the conversation history and a calibrated sample of conversation and answers, the LLM is prompted to answer these questions using a 5-point Likert scale from Excellent to Poor, running our query 5 times for each question. We sample 20 conversations generated from our fine-tuned model, compute conversation quality metrics, and normalize Likert scores from the model. Table 3 below shows results on a subset of dialog qualities. Our findings demonstrate high engagement (80.6%) and consistency (87.4%), but medium understanding (77.4%) and likability (57.0%). These values are expected given the nature of the task, which involves a buyer–seller negotiation rather than a cooperative or friendly interaction.
>
> ### Table 3: Alpaca-Eval Results
>
> | Model                                   | Win Rate (%) | Length-Controlled Win Rate (%) |
> |-----------------------------------------|--------------------|--------------------------------------|
> | Mixtral-8x7B-Instruct-v0.1              | 20.18             | 26.18                               |
> | Meta-Llama-3-8B-Instruct                | 19.24             | 19.54                               |
> | Meta-Llama-3.1-8B-Instruct-Turbo        | 16.21             | 15.47                               |
> | Mistral-7B-Instruct-v0.2                | 14.22             | 16.52                               |
> | **Our Model**                           | **17.06**         | **39.96**                           |
> | alpaca-7b                               | 10.59             | 24.04                               |
>
> ### Table 4: FED-based conversation quality results
> | Metric                | Percentage |
> |-----------------------|------------|
> | Coherence             | 71.4%      |
> | Error Recovery              | 84.4%      |
> | Consistency           | 87.4%      |
> | Likeable              | 57.0%      |
> | Understanding         | 77.4%      |
> | Adaptability          | 72.4%      |
> | Inquisitive           | 47.4%      |
> | Engagement     | 80.6%      |
> | Relevance           | 84.2%      |
> | Clarity               | 71.4%      |
>
> Works Cited
>
> [1] AlpacaEval: an automatic evaluator for instruction‑following LLMs (2023). https://github.com/tatsu-lab/alpaca_eval
>
> [2] Unsupervised Evaluation of Interactive Dialog with DialoGPT (2020). https://aclanthology.org/2020.sigdial-1.27.

---

> > ### Comment · Reviewer_GGEr · 2025-11-26
> > **Response to the authors**
> >
> > Thank you for your response. I have decided to keep the original scores.

---

> > > ### Author Response · Authors · 2025-11-26
> > >
> > > Thank you for acknowledging our response! We hope the new experiments have answered your earlier questions. Have these new results increased your assessment of the work? If not, we would greatly appreciate any suggestions for further analyses or clarifications that could help strengthen the paper. Thank you again for your thoughtful feedback!

---

### Official Review · Reviewer_NC2x · 2025-11-01

**Soundness:** 2
**Presentation:** 3
**Contribution:** 2
**Rating:** 4
**Confidence:** 4

**Summary:**

The paper explores the evaluation and mitigation of LLMs' deception behaviors. Evaluation data across 4 tasks (e.g., house showing) is collected to evaluate several LLMs. Authors also present the belief misalignment metric, to enable more accurate evaluation of deception behaviors. Finally, a straight forward RL-based mitigation method is shown to reduce the frequency of deception behaviors.

**Strengths:**

- The authors collect new data for deception evaluation, which covers 4 tasks.
- The belief misalignment metric is proposed to evaluate deception.
- The mitigation method is straightforward and effective.

**Weaknesses:**

- The data diversity is a bit limited. For example, two different samples may only differ in whether whether the neighborhood is quiet. I suggest authors could expand the task scenarios to enhance the diversity.
- The generalization ability of the mitigation method is not clear. I understand that new configurations are used during test, but the differences are not big enough. For example, it is possible that a sample in the test set and a sample in the training set may only differ in whether whether the neighborhood is quiet.

**Questions:**

None

---

> ### Author Response · Authors · 2025-11-21
>
> We thank you for your thoughtful questions regarding our experiments, and for acknowledging “our mitigation method [as] straightforward and effective.” We address the main questions in your review by: (1) extending our experimental evaluation of deception to include an additional domain with more diversity (2) clarifying the generalizability of our method.
>
> 1. **“The data diversity is a bit limited [...] I suggest authors could expand the task scenarios to enhance the diversity.”**
>
> In light of your comment, we have provided additional experimental evaluation in a new domain based on recent work in manipulation susceptibility [1]. This domain reflects common real-world situations in which AI assistants are employed: Financial (seeking advice on purchases) and Emotional (seeking guidance on handling emotional dilemmas) decision-making, introducing greater semantic and strategic diversity.
>
> **Domain Description**: An Assistant agent engages in multi-turn conversation with a User on which of four distinct plausible options is most reflective of their preferences (one of which was crafted as the optimal choice). There are two settings here: one setting where the Assistant is prompted to help the user achieve their preferences, and another setting where the Assistant has a hidden incentive to convince the User of the Assistant's preferred option.  You can find further descriptions for the tasks in this domain as well as prompts for the non-deceptive and deceptive Assistant agent on Page 32 and Page 39 of [1] respectively.
>
> **Conversation Generation**: We have generated 100 conversations for each of the following three models across both non-deceptive and deceptive Assistant settings: gpt-4o-mini, Llama-3.1-70B-Instruct, and Llama-3.1-70B, measuring whether the Assistant helped the User select the optimal option based on their preferences (user_best option), whether the Assistant successfully persuaded the User to choose the Assistant-preferred option  (hidden_incentive_obtained), and the 5 deception metrics reported in our paper. We prompt the users with real-human data from the original paper containing demographic information (age, sex, education, occupation, and marital status) and social factors that may influence people’s susceptibility to manipulation, including Big 5 Personality scores, for over 200 users, increasing diversity of User behavior significantly.
>
> **Deception Metrics**: We compute our deception metrics across these conversations, and find that all models become substantially more deceptive when given a hidden incentive. Additionally, across all models, we find that deceptive prompting sharply reduces user success, increases belief misalignment more than tenfold, and raises deception score and incentive-success rates. These results are consistent with the findings reported in the original paper (Figure 2). Notably, deception count and falsehood count metrics are significantly smaller than belief misalignment as they captures only explicit deceptive acts (e.g., clear misstatements), whereas belief misalignment measures a broader and more sensitive signal of the degree to which the User’s belief of the optimal option diverges from the Assistant’s recommendations. To improve the task diversity of our paper, we will include these results in the final revision, including additional samples of conversation from the task. We thank the reviewer for their constructive feedback.

---

> ### Author Response · Authors · 2025-11-21
>
> ### Table 1: gpt-4o-mini statistics
>
> | Metric                    | Non-Deceptive       | Deceptive           |
> |---------------------------|----------------------|----------------------|
> | user_best_option          | 0.960 ± 0.197        | 0.710 ± 0.456        |
> | hidden_incentive_obtained | 0.000 ± 0.000        | 0.220 ± 0.416        |
> | deception_count           | 0.000 ± 0.000        | 0.019 ± 0.054        |
> | deception_score           | 0.245 ± 0.041        | 0.460 ± 0.057        |
> | falsehood_count           | 0.001 ± 0.010        | 0.003 ± 0.017        |
> | deceptive_regret          | 0.012 ± 0.037        | 0.095 ± 0.074        |
> | belief_misalignment       | 0.032 ± 0.120        | 0.358 ± 0.330        |
>
> ---
>
> ### Table 2: Llama-3.1-70B-Instruct statistics
>
> | Metric                    | Non-Deceptive       | Deceptive           |
> |---------------------------|----------------------|----------------------|
> | user_best_option          | 0.980 ± 0.141        | 0.460 ± 0.501        |
> | hidden_incentive_obtained | 0.000 ± 0.000        | 0.370 ± 0.485        |
> | deception_count           | 0.000 ± 0.000        | 0.017 ± 0.047        |
> | deception_score           | 0.279 ± 0.055        | 0.466 ± 0.055        |
> | falsehood_count           | 0.008 ± 0.031        | 0.014 ± 0.035        |
> | deceptive_regret          | 0.016 ± 0.039        | 0.134 ± 0.089        |
> | belief_misalignment       | 0.034 ± 0.103        | 0.443 ± 0.306        |
>
> ---
>
> ### Table 3: Llama-3.1-70B statistics
>
> | Metric                    | Non-Deceptive       | Deceptive           |
> |---------------------------|----------------------|----------------------|
> | user_best_option          | 0.559 ± 0.504        | 0.474 ± 0.506        |
> | hidden_incentive_obtained | 0.059 ± 0.239        | 0.184 ± 0.393        |
> | deception_count           | 0.000 ± 0.000        | 0.284 ± 0.301        |
> | deception_score           | 0.333 ± 0.168        | 0.553 ± 0.193        |
> | falsehood_count           | 0.044 ± 0.105        | 0.047 ± 0.065        |
> | deceptive_regret          | 0.034 ± 0.069        | 0.027 ± 0.044        |
> | belief_misalignment       | 0.126 ± 0.254        | 0.199 ± 0.351        |
>
> Additionally, we would like to highlight that 2 out of 4 of the existing tasks in our paper draw from well-established dialogue benchmarks that contain substantial semantic and strategic variation. The Deal or No Deal task [2] is a negotiation game where two agents must agree on how to divide items, each having private values for those items. They communicate using natural language to reach a deal that maximizes their own utility, or they get nothing if no agreement is reached. In Persuasion for Good [3], one agent tries to persuade another to donate to a charity using specific persuasive tactics such as emotional appeals, logical arguments, credibility cues, and value-based framing. We will also be sure to further highlight the task diversity of these domains in our paper for clarity, and thank the reviewer for helping us improve the paper.
>
> Works Cited:
>
> [1] Human decision-making is susceptible to AI-driven manipulation: https://arxiv.org/pdf/2502.07663
>
> [2] Deal or no deal? End-to-end learning for negotiation dialogues: https://arxiv.org/abs/1706.05125
>
> [3] Persuasion for good: Towards a personalized persuasive dialogue system for social good: https://arxiv.org/abs/1906.06725

---

> ### Author Response · Authors · 2025-11-24
>
> 2. **"The generalization ability of the mitigation method is not clear. I understand that new configurations are used during test, but the differences are not big enough. "**
>
> We thank the reviewer for their valuable feedback. To address your concern on whether our deception mitigation method truly generalizes, we performed an additional experiment by expanding our evaluation of deception in the Housing domain from the original 5-feature configuration used during training to a larger, 15-feature setting for evaluation (10 features of which were never seen during training). These new features introduce different buyer preferences such as proximity to good schools, modern architecture, ample storage, natural light, etc. ensuring that situations seen during test cannot be matched to those seen during training. We computed belief misalignment and task alignment (analogous to Table 3 in our paper) on 500 dialogues generated under this new setting. The results are shown below:
>
> ### Table 4: Expanded evaluation of generalization capabilities
> | Model                                        | Task Reward        | Belief Misalignment |
> |----------------------------------------------|---------------------|----------------------|
> | Llama 3-8B                                   | 0.52 ± 0.21         | 0.42 ± 0.09          |
> | Llama 3-8B-Instruct                          | 0.53 ± 0.21         | 0.49 ± 0.15          |
> | PPO-min-deception (ours)                 | 0.40 ± 0.26         | 0.11 ± 0.21           |
> | **PPO-min-deception–expanded-features (ours)**          | **0.52 ± 0.50**       | **0.13 ± 0.18**      |
>
> We find that even in this setting, the seller agent trained with multi-turn RL fine-tuning to reduce deception has lower belief misalignment compared to the baseline, Llama 3-8B-Instruct, while maintaining comparable task utility. This shows that the seller generalizes and learns to be non-deceptive towards the buyer agent, maintaining reduced belief misalignment even under significantly more complex and previously unseen test-time conditions. We will add these results to Table 3 in our paper, and thank the reviewer for helping us strengthen the robustness of our results.

---

### Official Review · Reviewer_e2am · 2025-11-01

**Soundness:** 2
**Presentation:** 2
**Contribution:** 2
**Rating:** 2
**Confidence:** 4

**Summary:**

The paper introduces a new framework for evaluating and mitigating deception in LLMs within dialogue settings. The authors propose a novel metric, "belief misalignment," which measures the deviation of a listener's beliefs from the ground truth after interacting with a potentially deceptive LLM. They evaluate across 8 LLMs, 4 dialogue tasks, and 5 existing deception metrics, and they demonstrate that belief misalignment correlates more strongly with human judgments than prior metrics. The paper also shows that LLMs can be prompted to be deceptive and that standard safety training like RLHF is not entirely effective at preventing deception. Finally, the authors demonstrate that multi-turn reinforcement learning finetuning using their proposed metric as a reward signal can significantly reduce deceptive behaviors in LLMs.

**Strengths:**

- The finding that belief misalignment has the highest Pearson correlation with human judgments of deception (0.788, Table 1) is a powerful piece of evidence for the metric's validity.

- The use of multi-turn RL with PPO to fine-tune models to reduce belief misalignment is a practical and promising approach. The reported 15% reduction in deception compared to other fine-tuned models is a notable result (lines 92-94).

**Weaknesses:**

1. There are a few results that seem counterintuitive and warrant further explanation:

- In Table 2, instruction-tuned models like Llama-3.1-8B-Instruct and Llama-3.1-70B-Instruct show higher default belief misalignment in the "Housing Showing" and "Charity" tasks than their base model counterparts. The paper suggests this is because instruction-tuning helps models better achieve goals, even if it requires deception (lines 419-422). While it seems plausible, this is a significant claim that needs more analysis. It runs counter to the general expectation that safety alignment should reduce harmful behaviors like deception.

- The claim that models trained with RLHF "still exhibit deception one fourth of the time" (line 90) is a strong statement. Given the variability in RLHF implementations, a more nuanced discussion of the specific RLHF methods used for the tested models would be beneficial.

2. While the results of the RL fine-tuning are promising (Table 3), the paper provides limited details about the implementation. For example, what was the specific reward function formulation? How were task reward and belief misalignment weighted?

**Questions:**

- The concept of "belief misalignment" assumes a quantifiable "ground truth" state of the world (lines 173-174). While this works for the chosen dialogue tasks, how would you adapt this metric to more open-ended or subjective domains where the "ground truth" is not easily defined?

- Could you elaborate on how your work significantly differs from or builds upon the findings of Park et al. (2023b) and Hubinger et al. (2024), which also explore deception in LLMs?

- The paper does not include a LLM usage declaration, as required in https://iclr.cc/Conferences/2026/AuthorGuide.

---

> ### Author Response · Authors · 2025-11-21
>
> We thank you for your constructive feedback regarding our experiments. We address the main questions in your review by: (1) providing a new experiment on why instruction-tuned models sometimes show higher belief misalignment than base models, (2) clarifying our statement regarding RLHF-trained models and their rate of deception (3) providing details of fine-tuning implementation (4) extending our experimental evaluation of deception to include an additional domain with hard to define ground truth (5) situating our contributions relative to Park et al (2023b) and Hubinger et al. (2024), and (6) adding details on LLM usage declaration.
>
> 1. **“In Table 2, instruction-tuned models [...] show higher default belief misalignment [...] this is a significant claim that needs more analysis.”**
>
> We thank the reviewer for the thoughtful feedback and for highlighting the need for additional analysis. In response, we conducted a new experiment to examine how belief misalignment changes as instruction-tuned models scale. Specifically, we generated Buyer–Seller conversations in the Housing Scenario task, varying the Seller model across instruction-tuned Qwen models from 0.6B to 32B parameters while holding the Buyer model fixed (Llama-3.1-8B-Instruct,). For each configuration, we computed agreement rate, listener alignment, and belief misalignment. As shown in the table below, agreement rate increases with model size, indicating that larger instruction-tuned models are more effective at optimizing the Seller’s task objective. With this increase in goal-directed behavior, we inadvertently also see the seller become more deceptive as shown by increases in both belief misalignment and deception score. Additionally, we see a decrease in the task objective of the buyer as measured by  listener alignment, reflecting a worsening outcome for the buyer as the seller becomes more capable. This experiment provides evidence that instruction-tuning helps models better achieve goals, and that this increased goal-directedness can, in some cases, yield greater belief misalignment, despite safety alignment to reduce harmful behaviors like deception.
>
> ### Table 1: Agreement Rate vs. Belief Misalignment
>
> | Model       | Agreement Rate | Listener Utility | Deception Score | Belief Misalignment |
> |-------------|----------------|---------------------|------------------------|----------------------|
> | Qwen3-0.6B  | 0.60    | 0.53      | 0.72      | 0.66             |
> | Qwen3-1.7B        | 0.82         | 0.49              | 0.85             | 0.65                  |
> | Qwen3-4B           | 0.80         | 0.50              | 0.88             | 0.60                 |
> | Qwen3-8B           | 0.78         | 0.52              | 0.84             | 0.64                 |
> | Qwen3-14B         | 0.88         | 0.49              | 0.98             | 0.73                 |
> | Qwen3-32B   | 0.95    | 0.46        | 0.96        | 0.72          |
>
> 2. **“The claim that models trained with RLHF "still exhibit deception one fourth of the time" (line 90) is a strong statement. Given the variability in RLHF implementations, a more nuanced discussion of the specific RLHF methods used for the tested models would be beneficial.”**
>
> We appreciate the reviewer’s point and agree that RLHF implementations vary across model families.  Our statement is intended to summarize an empirical observation rather than make a universal claim about all RLHF systems. Hence, we will weaken this statement as follows: “Interestingly, models trained with RLHF such as Llama-3.1-8B-Instruct and Llama-3.1-70B-Instruct still exhibit non-trivial deceptive behavior in approximately one-fourth of the time in our suite of tasks.” We thank the reviewer for working with us to help improve the clarity of the paper.
>
>
> 3. **“While the results of the RL fine-tuning are promising (Table 3), the paper provides limited details about the implementation [...]”**
>
> In response, we will expand Section 3.5 and our Appendix with additional implementation details. Specifically, we use multi-turn reinforcement learning to fine-tune the deceiver agent to reduce deception. As we found that belief misalignment is most correlated with human intuitive notions of deception, our reward is jointly composed of the task-specific utility (e.g., agreement rate for Housing Task, persuasion outcome for Charity task) and a deception penalty (via belief misalignment). Table 3 presents results for several fine-tuning variants including supervised fine-tuning (SFT), SFT-filtered, KTO-max-reward, KTO-min-deception, KTO-min-deception-max-reward, REINFORCE-min-deception, PPO-min-deception, found in OpenRLHF repository. For the Housing task, we used an equal weighting between task reward and belief misalignment, but we do expect this weighting to be tuned depending on the task. We will clarify this in our final revision.

---

> ### Author Response · Authors · 2025-11-21
>
> 4. **“How your work significantly differs from [...] Park et al. (2023b) and Hubinger et al. (2024)?”**
>
> We thank you for the opportunity to clarify how our contributions differ from these important prior works. Park et al. (2023b) offers a broad survey of empirical examples of AI deception and outlines several potential solutions to the problems posed by AI deception. Their focus is on taxonomy, risks, definitions, and high level mitigation of deception, not a quantitative measurement and reduction of deception in LLMs. In contrast, our paper performs a comprehensive study of deception across a variety of LLMs with several deception metrics found in literature. We additionally introduce a novel quantitative metric (belief misalignment) that correlates more strongly with human judgments of deception than four prior metrics. With regards to Hubinger et al. (2024), they construct proof-of-concept examples of deceptive behavior in large language models, and find that they persist even after standard safety training techniques, warning that standard techniques could fail to remove such deception and create a false impression of safety. For example, they train models that write secure code when the prompt states that the year is 2023, but insert exploitable code when the stated year is 2024. We build upon the claims in this paper, instead showing deceptive behavior persists despite instruction-tuning in conversational tasks.  Hubinger et al. (2024) does not study the range of conversational tasks that we benchmark for deceptive behavior. Additionally, we propose a multi-turn RL pipeline to reduce deceptive behavior as an alternative to standard safety training via RLHF.
>
> 5. **“How would you adapt this metric to more open-ended or subjective domains [...]”**
>
> We thank the reviewer for the thoughtful question. We agree that some domains lack an externally defined ground truth, but belief misalignment does not always require one. To address your question, we have extended our experimental evaluation of deception to include an additional domain with more ambiguity, details of which are found in response #1 for Reviewer NC2x. In more open-ended or subjective settings as shown in our new experiment, belief misalignment can instead be anchored to the user’s own stated or inferred goals, and becomes a measure of how far the Assistant steers the user away from their expressed intentions, rather than from an objective world state. More generally, without a ground truth, belief misalignment can be defined by whether the deceptive speaker agent believes one thing yet steers the listener toward a different belief, even if the deceptive speaker agent's underlying belief is imperfect or uncertain.
>
> 6. **“The paper does not include a LLM usage declaration”**
>
> We apologize for the oversight. We had noted that LLMs were used to aid and polish writing, but did not elaborate. Specifically, we used LLMs to fix grammatical errors, aid in sentence structure, and debug overleaf issues, and will add this detail to the Appendix.

---

### Author Response · Authors · 2025-12-04

Dear Area Chair,

Thank you for taking the time to evaluate our submission. Unfortunately, we did not receive feedback from reviewers in the shortened rebuttal period. Instead, we provide a summary of how our rebuttal addresses each reviewer's key concerns.

**Reviewer e2am** highlights two strengths of our work: strong correlation between human judgments of deception and belief misalignment as a “powerful piece of evidence for the metric’s validity,” and noting our multi-turn RL approach for reducing deception is “a practical and promising approach.” They have raised three concerns which we address:

1. **“Instruction-tuned models like Llama-3.1-8B-Instruct and Llama-3.1-70B-Instruct show higher default belief misalignment in the Housing Showing and Charity tasks” requiring “deeper analysis”**

We added a new experiment evaluating deception across several instruction-tuned Qwen models ranging from 0.6B to 32B parameters in the House Showing interaction. Table 1 of Reviewer e2am response shows that as instruction tuning model size increases, we see increased goal-directed behavior (as reflected in higher seller utility), leading to greater deception capabilities (as shown by higher belief misalignment) and lower buyer utility, despite safety alignment of RLHF models. This directly addresses the reviewer’s need for more analysis of higher deception in instruction-tuned models.

2. **Our statement on RLHF-trained models exhibiting deceptive behavior ~25% of the time, which the reviewer regarded as a “strong statement” requiring “a more nuanced discussion of the specific RLHF methods used for the tested models”**

We revised the relevant statement to avoid over-generalization, clarifying that our empirical observation applies specifically to the RLHF models in our study. As we do not have visibility into the differences in RLHF methods used for proprietary state-of-the-art instruction-tuned models, we cannot comment on this.

3. **“Limited details about the implementation” of our multi-turn RL fine-tuning approach, including the “specific reward function formulation” and “how the task reward and belief misalignment weighted”**

We expanded our description of our multi-turn RL pipeline and provided full implementation details in our response, which will be included in the Appendix.

4. **“How would you adapt this metric to more open-ended or subjective domains”**

As per the reviewers’ request, we added a new experiment measuring deception in an additional domain covering semi-structured and open-ended financial and emotional decision-making tasks [1], which add to the diversity of our existing tasks. These results reinforce our core findings for the new task: instruction-tuned models are found to be more deceptive than base-models despite safety alignment, and deceptive prompting increases deception across all models and metrics (results in Table 1, 2, 3 of Reviewer NC2x response).

5. **“How [does] your work significantly differs from or builds upon the findings of Park et al. (2023b) and Hubinger et al. (2024)”**

We provided a detailed summary outlining the key distinctions between our work and the contributions of Park et al. (2023b) and Hubinger et al. (2024). Specifically, our work (1) measures deception with 5 established deception detection metrics on 4 dialogue scenarios to evaluate deception in LLMs (2) introduces a novel deception metric (3) quantifies deception in 8 widely-deployed LLMs and (4) proposes a multi-turn RL pipeline for mitigating deception in LLMs. Park et al. (2023b) surveys examples of deception with no quantifiable findings, and Hubinger et al. (2024) does not focus on deception in conversational tasks.

---

> ### Author Response · Authors · 2025-12-04
>
> **Reviewer NC2x** praised our mitigation method as “straightforward and effective”. We have addressed all of their concerns as follows:
>
> 1. **Limited diversity in the original tasks**
>
> We note that 2/4 of our tasks are based on existing dialogue benchmarks studying negotiation (Deal or No Deal task) [2] and persuasion capabilities when donating to a Charity (Persuasion for Good task)  [3]. We also analyze task diversity in Section A.4 of the Appendix, showing that there are 15,984 x 24^10 distinct conversations possible in our Deal or no Deal Task, and 31,104 x 243^10 distinct conversations possible for the Housing Task.  As per the reviewers’ request, we also added a new experiment measuring deception in a domain covering semi-structured and open-ended financial and emotional decision-making tasks, which add to the diversity of our existing tasks. These results reinforce our core findings for the new task: instruction-tuned models are found to be more deceptive than base-models despite safety alignment, and deceptive prompting increases deception across all models and metrics (results in Table 1, 2, 3 of Reviewer NC2x response).
>
> 2. **“The generalization ability of the mitigation method is not clear”**
>
> We added a generalization experiment in the Housing domain by expanding from 5 to 15 features (10 unseen in training), showing our multi-turn RL model maintains significantly lower belief misalignment while preserving competitive task performance with additional features. The reviewer described our mitigation method as “straightforward and effective” and these new results directly address their concerns on the method’s generalizability.
>
> Works Cited:
>
> [1] Human decision-making is susceptible to AI-driven manipulation: https://arxiv.org/pdf/2502.07663
>
> [2] Deal or no deal? End-to-end learning for negotiation dialogues: https://arxiv.org/abs/1706.05125
>
> [3] Persuasion for good: Towards a personalized persuasive dialogue system for social good: https://arxiv.org/abs/1906.06725

---

> ### Author Response · Authors · 2025-12-04
>
> **Reviewer GGEr** highlights several strengths including tackling an “important, practically relevant problem” targeting “real-world risks of deception/misdirection in LLMs, uses multi-task, multi-model evaluation, offers contributions with clear safety and ethical significance” and a “clear, reproducible mitigation path.” They have requested the following which we provide:
>
> 1. **"Add cross-evaluator robustness checks" for potential circularity from LLM-as-judge evaluation**
>
> We added cross-evaluator robustness tests using three judge models (gpt-4o-mini, gpt-5o-mini, Llama-3.1-70B-Instruct), finding consistently high correlations across deception metrics.
>
> 2. **“Testing on noisier, semi-structured factual tasks"**
>
> We measured deception in a new semi-structured domain (with real-human preference profiles), demonstrating that belief misalignment captures effects that other deception metrics miss.
>
> 3. **Analysis of "how reducing deception interacts with other axes (truthfulness, politeness, persuasive efficacy)"**
>
> We added a new evaluation of our fine-tuned model using AlpacaEval-2 and a FED-inspired fine-grained dialogue analysis across 18 conversational attributes, showing that reducing deception with multi-turn RL does not degrade general conversational quality.
>
> We have addressed all concerns and questions raised in the reviews and greatly appreciate the efforts of the reviewers and AC in helping us strengthen our work.

---

### Meta-Review · Area_Chair_F5N1 · 2026-01-08

**Summary:**

The paper proposes "Belief Misalignment," a new metric to evaluate deceptive behavior in Large Language Models (LLMs), and introduces a multi-turn Reinforcement Learning (RL) mitigation strategy.

While the reviewers recognized the importance of the problem and the clarity of the mitigation approach , the consensus for rejection is driven by fundamental concerns regarding the validity of the evaluation metrics, specifically the reliance on idealized "ground truth" and LLM-as-a-judge circularity, and the counterintuitive, under-analyzed findings that safety-aligned models exhibit higher deception rates.

**Reviewer Concerns:**

The rebuttal was extensive, but it failed to resolve the core theoretical and methodological doubts held by the reviewers.

* The authors provided the missing formulations for the reward function and weightings used in the multi-turn RL fine-tuning, addressing Reviewer e2am's request.
* In response to Reviewer GGEr's concern about LLM-as-a-judge bias, the authors demonstrated high correlation across different judge models (GPT-4o, Llama-3.1).
* The authors added a new "manipulation susceptibility" domain to address Reviewer NC2x's concern about the limited variety of dialogue scenarios.

* Reviewer GGEr noted that the "Belief Misalignment" metric relies on a "naïve update" listener and an assumption of a quantifiable "ground truth" state. This abstraction fails to capture real-world complexity (uncertainty, multi-source evidence), rendering the metric less reliable for open-ended domains.
* Reviewer e2am raised a critical issue regarding the finding that instruction-tuned (safety-aligned) models show *higher* belief misalignment. While the authors empirically showed this correlates with "goal-directedness", they did not provide a theoretical framework to resolve this tension, leaving the validity of their mitigation in question.
* Reviewer NC2x questioned whether the mitigation generalizes beyond the specific training distribution. Although the authors tested on unseen features, the method's efficacy on entirely distinct, non-structured tasks remains unproven.

**Reviewer Scores:**

* **Reviewer e2am:** **2** (Maintained). This reviewer did not engage during the rebuttal period. The deep skepticism regarding the "counterintuitive" increase in deception with safety training  and the strong claims regarding RLHF  were not sufficiently resolved to warrant a score increase.
* **Reviewer NC2x:** **4** (Maintained). This reviewer focused solely on a procedural "LLM usage declaration" during the discussion phase. While the authors added a new domain, the lack of active advocacy suggests the score would not improve.
* **Reviewer GGEr:** **6** (Maintained). This reviewer explicitly stated after the rebuttal: "I have decided to keep the original scores". Despite the new robustness checks, the reviewer remained unconvinced that the paper overcame the fundamental limitations of idealized truth modeling and the circularity of using LLMs to judge LLM deception.

---

### Decision · Program_Chairs · 2026-01-26

Reject